# Palliative Care Landscape in the COVID-19 Era: Bibliometric Analysis of Global Research

**DOI:** 10.3390/healthcare10071344

**Published:** 2022-07-20

**Authors:** Hammoda Abu-Odah, Jingjing Su, Mian Wang, Sin-Yi (Rose) Lin, Jonathan Bayuo, Salihu Sabiu Musa, Alex Molassiotis

**Affiliations:** 1School of Nursing, The Hong Kong Polytechnic University, Hong Kong 999077, China; mian.wang@connect.polyu.hk (M.W.); jonathan.bayuo@connect.polyu.hk (J.B.); alex.molasiotis@polyu.edu.hk (A.M.); 2Centre for Advancing Patient Health Outcomes, A JBI Affiliated Group, School of Nursing, The Hong Kong Polytechnic University, Hong Kong 999077, China; 3Nursing and Health Sciences Department, University College of Applied Sciences (UCAS), Gaza P860, Palestine; 4WHO Collaborating Centre for Community Health Services (WHOCC), School of Nursing, The Hong Kong Polytechnic University, Hong Kong 999077, China; 5School of Nursing, Li Ka Shing Faculty of Medicine, The University of Hong Kong, Hong Kong 999077, China; rosie@connect.hku.hk; 6Department of Applied Mathematics, Hong Kong Polytechnic University, Hong Kong 999077, China; salihu-sabiu.musa@connect.polyu.hk; 7Operational Research Center in Healthcare, Near East University TRNC, Mersin 10, Nicosia 99138, Turkey

**Keywords:** palliative care, COVID-19, bibliometric analysis

## Abstract

Despite the increasing number of publications globally, the COVID-19 pandemic has underscored significant research gaps that should be resolved, including within PC-related research. This study aimed to map and understand the global trends in palliative care (PC)-related COVID-19 research and provide quantitative evidence to guide future studies. We systematically searched four databases between 1st January 2020 and 25th April 2022. The VOSviewer, Gephi, and R software were utilized for data analysis and results visualization. A total of 673 articles were identified from the databases between 1st January 2020 and 25th April 2022. Canada (6.2%), Australia (5.4%), and the United Kingdom (3.8%) were the most productive countries regarding articles published per million confirmed COVID-19 cases. A lack of international collaborations and an uneven research focus on PC across countries with different pandemic trajectories was observed. The PC research in question focused on cancer, telehealth, death and dying, and bereavement. This study’s conclusions support the recommendation for international collaboration to facilitate knowledge and practice transformation to support countries with unmet PC needs during the pandemic. Further studies are required on the grief and bereavement support of families, healthcare professionals and patients with other life-threatening illnesses.

## 1. Introduction

The coronavirus 2019 (COVID-19) pandemic has become a global public health concern [1]. It has negatively impacted social, economic, and healthcare systems globally [2]. The quantity and quality of health services provided to patients have been impacted considerably by the high incidence of infections [3]. Existing disparities in access to health services are predicted to surge as the gap widens between countries with and without adequate resources and funding allocations to overcome the effects of the pandemic on delivering care to patients with life-threatening illnesses [2,4].

Palliative care (PC) has played a pivotal role in the response to COVID-19 by providing symptom management and physical and psychological support to patients and families [5]. The COVID-19 pandemic has forced national and international PC institutions to produce guidance statements for PC specialists to confirm the safety and optimal care for people dying from COVID-19 [6,7]. This has stimulated researchers to conduct studies that have led to a plethora of health-related publications [8]. These publications cover a wide range of subject matter, such as COVID-19 transmission modes, the determinants of outcomes, and the PC support applicable to persons living with COVID-19 and their families. A quick search in the World Health Organization (WHO) portal, as of 20th April 2022, shows up to 466,114 COVID-19-related publications in just 27 months since the start of the pandemic [9]. Despite the increasing number of publications, the pandemic has underscored significant research gaps that should be resolved, including within PC-related research [4,10]. The first COVID-19 bibliometric mapping was conducted by Chahrour et al. [11], and it targeted only PubMed and WHO database publications between December 2019 and March 2020. A total of 564 articles were identified from 39 countries, and China ranked the highest among these in terms of the number of publications, with 67% of the total number of publications. While the recent bibliometric mapping review conducted by Abu-Odah et al. [10] focused on global PC between 2002 and 2020, this was not specific to COVID-19 and was restricted to the study of articles published in English. The study results showed that the USA and the UK were the two most productive countries in regard to PC-related publications. Despite the important findings of these two reviews, there no bibliometric study has shed light on the global developments in PC-related COVID-19 research. To address this gap, we conducted a bibliometric network analysis focusing on the PC-related COVID-19 research. It is argued that the quantitative findings of this study will deliver significant insight for future directions in the PC-related COVID-19 research.

## 2. Methods

### 2.1. Search Strategy

A systematic literature review with bibliometric analysis of PC-related COVID-19 research was performed from 1st January 2020 to 15th August 2021, and was updated on 25th April 2022, using metadata extracted from the Medline (EbscoHost), SCOPUS (Elsevier), Cumulative Index of Nursing and Allied Health Literature (CINAHL) (EbscoHost), and Web of Science (Thomson Reuter) databases. The rationale for using these databases is that they are the most commonly comprehensive databases that encompass the biomedical and social sciences literature [12].

We retrieved the literature from the aforementioned databases using the keywords adopted in previous review studies [10,13,14,15,16]. The “palliative care” term with its alternative search keywords (“palliative medicine” OR “hospice care” OR “terminal care” OR “end-of-life care” OR “end of life care” OR “palliat*” OR “life-limiting” OR “life-threatening” OR “incurable disease” OR “supportive care”) was searched. The COVID-19 term with its alternative search keywords (“novel coronavirus 2019” OR “2019-nCov” OR “2019 Novel Coronavirus” OR “coronavirus” OR “middle east respiratory syndrome” OR “coronavirus disease 2019” OR “coronavirus 2019” OR “COVID 2019” OR “COVID 19” OR “nCOV” OR “SARS” OR “MERS” OR “SARS-CoV-2” OR “COVID-19” OR “COVID*” was also searched. Both terms were combined with the Boolean operator “AND” to generate specific results related to PC-related COVID-19. We screened the title and abstract of each publication to identify relevant papers. We included all articles focusing on PC-related COVID-19 publications, regardless of language. The search process of the database is presented in Appendix A.

### 2.2. Inclusion Criteria

The inclusion criteria were as follows:Original articles only.Articles stating the above-mentioned keywords in the title and abstract.Published between 1 January 2020 and 25 April 2022.Articles written in any language.The exclusion criteria were as follows:PC papers unrelated to the pandemic.Reviews, editorial notes, conference abstracts, letters, discussion, and erratum.

### 2.3. Screening and Data Extraction

Two of the authors (H.A.O. and JJ.U.) independently screened the title and abstract of each paper from the above-mentioned databases. Those papers that did not meet the inclusion criteria were excluded after an agreement between the authors. A third author (J.B.) was responsible for making the final decision of any uncertainly that the first two authors encountered during the assessment of the papers. The eligible articles were then exported into the “Comma-Separated Values” file and this was subsequently used to delete duplicates. The flow diagram of included articles and the reasons for excluding articles are reported in Figure 1.

### 2.4. Data Cleaning and Adjustment

The CSV metadata file of each database was slightly inconsistent with those in the other databases; thus, we undertook several steps before analyzing the bibliometric metadata to avoid duplication and missing any data. First, all of the databases’ files were modified to adhere to the Scopus file in terms of column titles. After modification and the exclusion of unnecessary columns, the files were combined into a single CSV file. The duplicated articles were removed based on each paper’s digital object identifier. The final bibliometric data were adjusted and corrected, including countries’ names, institutions, journals, and references, and then were transferred to the software for analysis.

### 2.5. Data Analysis and Visualization

Descriptive statistical analyses, including frequencies and percentages, were applied to identify the dynamic trends in the publications in the PC-related COVID-19 field. Data analysis was conducted utilizing the software R, version 4.0.3. The “bibliometrix” package, an R-tool for science-mapping analysis, was used to analyze and visualize PC-related COVID-19 research outputs [17]. We also utilized VOSviewer (version 1.6.13) [18] and Gephi (version 0.0.2) [19] in mapping the findings. VOSviewer is a free software that allows researchers to construct and visualize maps/networks easily with a high quality and high resolution. It was mainly utilized for cluster mapping and visualizing the co-authorship analyses of keywords and patterns of cooperation between institutions and countries by determining their frequency and total link strength. Despite the importance of VOSviewer, it only displays the nodes in a bibliometric network and does not present the edges between nodes. It also cannot be used for advanced analysis, such as of centrality and betweenness centrality, which are commonly used criteria for the analysis of co-authorship. Gephi is another free software for the visualization and analysis of large network graphs. It offers extensive visualization capabilities, which make it possible to customize visualizations in more detail. It was utilized to run advanced analyses, such as of centrality and betweenness centrality to analyze co-authorship [20]. Gephi is less focused on network analysis than VOSviewer is.

## 3. Results

### 3.1. Characteristics of the Retrieved Articles

The initial search in the Scopus and WOS databases yielded 1910 papers. After removing duplicates and unrelated articles, 673 articles remained for final analysis and visualization (Figure 1). Of note, 246 articles were published in 2020, 323 were published in 2021, and a further 104 articles were published until the last search in 25 April 2022. The retrieved articles were produced by 5204 authors, with an average of 7.73 authors per article. At the time of data extraction, articles were cited 4987 times, with an average of 7.83 citations per article (Table 1).

### 3.2. Most Common Journals

The 673 remaining articles were published in 357 scholarly journals. In total, the top 10 journals published 213 papers, accounting for 31.6% of the total number of publications, which received 640 citations (Table 2). The most frequently cited journal was the *Journal of Pain and Symptom Management* (*n* = 60, 8.9%), followed by *Palliative Medicine* (*n* = 32, 4.6%), and the *American Journal of Hospice and Palliative Medicine* (*n* = 30, 4.5%). Half of the top 10 cited journals were from North America. Although the *Indian Journal of Palliative Care* ranked fifth in terms of the frequency of its publications, the h-index of this journal was relatively low (H_index = 3) compared to other journals that published fewer articles. The same applies for *Soins journal*, which published 11 papers but had fewer citations. Among the top 10 journals, only three journals had no impact factor (IF), while three journals had an IF of >3.5.

### 3.3. Country Collaborations

An analysis of geographic origin showed that the USA was the leading country with 183 (27.2%) articles, followed by the UK with 84 (12.5%) articles and India with 45 (6.7%) articles. The top 10 productive countries were stratified based on article numbers per million confirmed COVID-19 cases and deaths (Table 3). The findings showed that Canada was ranked first, followed by Australia and the UK. These results also revealed that the highest average number of citations was 10.39 for articles from the USA, followed by France (7.92) and Italy (7.41). The lowest average number of citations was reported in India (1.71). Although India ranked at the top in terms of the frequency of publications, its overall citation performance was low.

The country co-authorship analysis in Figure 2 shows that the USA (from North America) stands out as the top-ranked country concerning cooperation in research with other countries, followed by the UK (from Europe) and Australia (in Australasia). India was the key research center in South Asia, Saudi Arabia in the Middle East, and Uganda in Africa. Despite this cooperation, the strength of relations with other countries was weak. Strong relations were observed only in the USA, the UK, Australia, Italy, Canada, France, and Germany.

### 3.4. Institutions

A total of 1891 institutions published relevant articles; 34 institutions met the criterion of having a minimum of two documents. The Gephi software was utilized to identify the top leading institutions by calculating the hyperlink-induced topic search values. The findings underscored the minimal cooperative relationships between institutions (Figure 3). A robust international collaboration was observed between Sweden and the USA, and between the UK and Switzerland. Cooperation was also noted between Italy and Uganda. No cooperation, however, was noted among many institutions.

### 3.5. Keywords Co-Occurrence Analysis

Out of 1129 keywords extracted from the databases, 39 keywords met the criterion to have a minimum number of four occurrences. The final network consisted of 39 nodes and 630 relations, with an average of 16.1 relations between keywords. The 39 keywords were distributed into seven clusters representing seven main research themes. The most frequent keywords were concentrated in the first three clusters.

The first cluster (red) included 10 keywords and focused on the topics of cancer, psychological distress, anxiety and depression, distress, chemotherapy, radiotherapy, prognosis, treatment, COVID-19, and telemedicine.

The second cluster (green) included seven keywords and concentrated on the topics of palliative care, hospitalization, nursing home, dementia, adult, education, and qualitative research.

The third cluster (blue) included seven keywords and focused on the topics of death and dying, terminal care, home care, communication, quality of life, public health, and ethics.

The network visualization of author keywords showed that cancer, telehealth, death and dying, hospitalization, bereavement, communication, advance care planning, nursing, qualitative research, ethics, and quality of life are the most common areas that have received research attention. The keywords in Figure 4 are illustrated in Table 4 with their occurrence frequency and total link of strength.

## 4. Discussion

This bibliometric network analysis has provided a map of the global PC-related COVID-19 research, showing a rapid increase in PC-related COVID-19 research led by countries with a high number of COVID-19 cases. The USA was the most productive country in this area in terms of the frequency of their publications, with four of their institutions ranking among the top 10 positions. Concurrently, the most cited papers emerged from the UK, with a clinical focus on managing patients with COVID-19 in PC [21,22]. The increased productivity of publications in the USA might be related to the increasing number of infections and deaths from COVID-19 [23] and the long history of PC [24,25,26]. By 3rd September 2021, over 219 million infections and 4.5 million deaths were reported worldwide [23]. Most of these deaths were among older adults and patients with underlying, life-threatening ailments [27]. According to the WHO pandemic-monitoring data, the added disease burden of COVID-19 requires the urgent enhancement of the operation and implementation of palliative care in settings with pandemic outbreaks and a lack of development in PC services [28,29].

International collaboration in PC-related COVID-19 research was observed to be low, which resonates with the study findings of Abu-Odah et al., where the authors noted that from 2002 to 2020, collaboration in global PC research was low [10]. Despite the low rates of cooperation, the USA is still leading in international efforts, as underscored by this study. This finding is in line with that of a previous study [30]. International cooperation is a fundamental method for sharing health knowledge, enhancing care, and increasing research capacity [31]. Low levels of collaboration across countries might be attributed to the small number of published articles in many of these countries. Another explanation might be the novelty of COVID-19, containment policies, and health care system incompatibilities, which have obligated each country to follow its research priorities and context-specific needs. The shared global concern of the COVID-19 pandemic provides an avenue for cross-country and cross-continental collaboration to strengthen solidarity.

Most previous reviews have revealed that PC research has focused on cancer-related issues [10,32,33], as cancer patients are one of the largest groups of patients with life-threatening illnesses. Although the included studies also concentrated more on the PC of the cancer patient population, their foci were more on different outcomes of interest, including pandemic-related mental crises and added symptom burden post-COVID-19, and the accessibility of PC services and alternatives for this cohort were emphasized. PC is not limited to a specific disease group; it is a multidimensional approach to end-of-life care [34]. Scholars have attributed the focus of PC-related COVID-19 research to cancer patients, as they are a vulnerable group at risk of COVID-19-related infection [35,36]. For instance, the mortality rate of lung cancer patients ranges between 25 and 55%, compared with 10% in other COVID-19 patients [37,38]. The high prevalence of deaths among patients has led to a plethora of research focusing on death, dying, and bereavement-related research. Despite the increasing attention in this regard, less attention has been paid to persons with other chronic ailments who contract COVID-19, such as dementia patients. Further studies are warranted among these vulnerable patients.

Grief is a key issue attributed to the pandemic that affects patients, families, and healthcare professionals [39]. The pandemic has created obstacles for the support of grievers [28]. This may explain why grief and bereavement-related topics have been frequently discussed in several studies. Most previous studies have shed light on professionals and pediatric bereavement, with little attention given to families. Further work that is related to the families’ grief and bereavement is required. Integrating bereavement care into health care should also be considered to overcome the negative consequences of COVID-19 for families and professionals.

Health technologies were adopted as substitutional methods to improve access to health services to adhere to pandemic containment measures undertaken by countries that had limited their in-person healthcare delivery [40]. Utilizing telehealth technologies is not new and has become widely adopted in healthcare settings. However, the usage of these technologies is relatively low [40]. Before the pandemic, there was evidence showing interest in utilizing telehealth services [41,42]. During the pandemic, health policies initiatives have been made to facilitate telehealth access and to promote delivering specialty care rapidly [43]. As shown by this study, telehealth services have been used as a method of communication between health facilities and patients. Such technologies have helped in the continuity of healthcare delivery to patients and have improved patient health outcomes [44]. Adopting telehealth services can also minimize the risk of infection between healthcare professionals and patients with life-threatening alignments. Thus, the expansion of telehealth technologies in the future is needed due to its benefits; however, implementing telehealth requires adequate preparation and technical expertise in the healthcare setting [45]. Thus, training PC professionals is required to help them to use telehealth remote platforms to deliver services to patients and families.

This review did not limit the search to only articles written in the English language; however, English-speaking countries still ranked at the top in terms of productivity and number of citations. Although China was ranked in the second-top position with an outstanding contribution to COVID-19 research [30], it seemed it had made minimal contribution to the PC-related COVID-19 research. Most publications from China have focused on the causes of the virus, diagnosis and treatment strategies, prevention and control, and the use of traditional Chinese medicine [46,47]. Low numbers of publications in China, as well as in Germany and Spain, might also be attributed to the different operation and implementation of PC in their healthcare systems.

### Strengths and Limitations

Our bibliometric analysis of PC-related COVID-19 publications offers a global overview of the studies that have been conducted on this topic, which allows the identification of the research gaps for future research. The search conducted in this study was not restricted to articles written in English, making the findings more generalizable. Despite these strengths, some limitations are noteworthy. Firstly, the data were sourced from Scopus and WOS only, with no further manual searching. Furthermore, articles published in 2020 have received a high number of citations compared with the newly published articles which may not reflect the importance of the latter studies accurately. Finally, the duration of the search period was short, and the rate of publications on the topic of this pandemic and other citation parameters may need to be altered as we come to understand more about COVID-19 and patients and as health systems adjust to it and learn to cope more efficiently.

## 5. Conclusions

This bibliometric network analysis delivers an overview of PC-related COVID-19 studies, which often focus on cancer care and technology-based care delivery. International collaboration should be fostered to facilitate knowledge and transform practices to support countries with unmet PC needs during the pandemic. Further studies are required in grief and bereavement support for families and healthcare professionals as well as in patients with other life-threatening illnesses.

## Figures and Tables

**Figure 1 healthcare-10-01344-f001:**
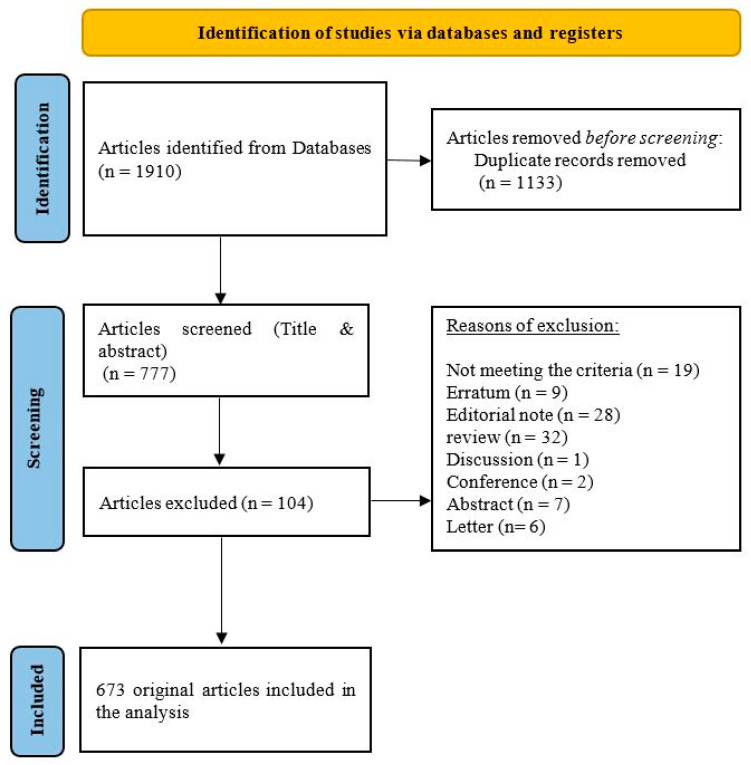
PRISMA flow diagram of the included articles.

**Figure 2 healthcare-10-01344-f002:**
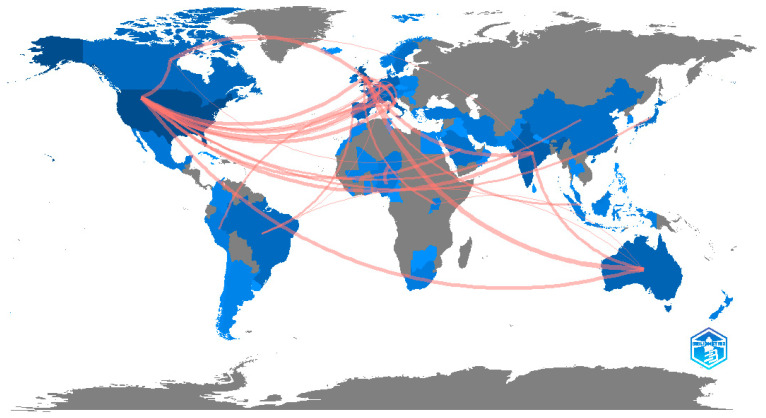
Network visualization map of cooperation between countries using “R”. Darker blue colors represent the most commonly collaborative countries regarding this topic, in terms of the frequency of their publications.

**Figure 3 healthcare-10-01344-f003:**
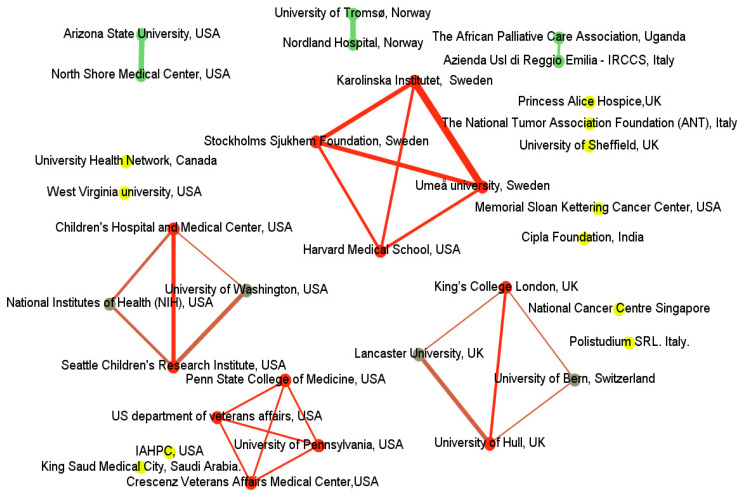
Network visualization map of cooperation across institutions created using Gephi. Nodes in the resultant network were recolored based on hyperlink-induced topic search value. Nodes with red color represent higher values that reflect highly influential institutions, followed by nodes with green and yellow. The thickness line between nodes also presents a strong relationship between institutions.

**Figure 4 healthcare-10-01344-f004:**
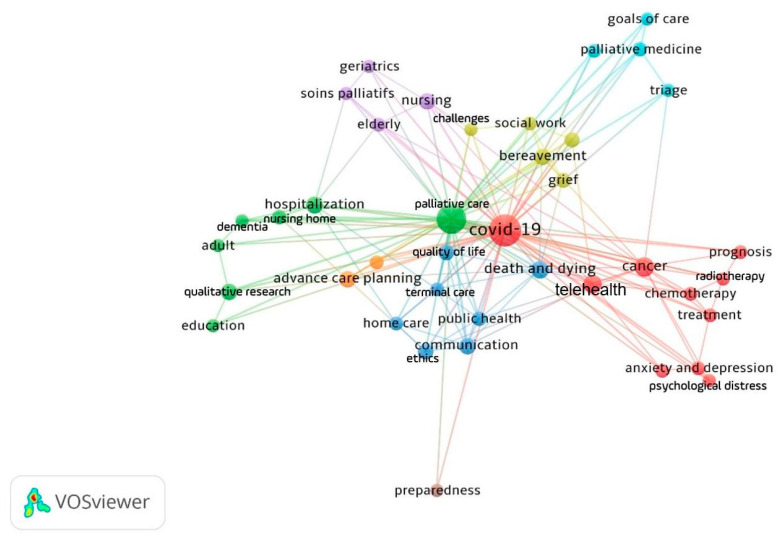
Network visualization map of most frequent author keywords created using VOSviewer. The colors in the map show seven clusters representing seven research themes. Nodes with similar colours represent a cluster of related terms. The figure was created using VOSviewer.

**Table 1 healthcare-10-01344-t001:** Characteristics of included articles (N = 673).

Characteristics	Results
**Main information about data**	
Sources (journals)	357
Average citations per documents	7.828
Average citations per year per	2.90
Number of references in included articles	4927
**Document contents**	
Keywords plus	2985
Author’s keywords in the paper	1129
**Authors**	
Total authors in included papers	5204
Authors in multi-authored documents	5171
**Authors collaboration**	
Single-authored documents	39
Documents per author	0.129
Authors per document	7.73
Co-authors per documents	10.4

**Table 2 healthcare-10-01344-t002:** Top 10 journals in the field of PC-related COVID-19 research.

Rank	Journals (JIF, Quartile (2020)) ^†^	Articles *n* (%)	Total Citations	H_Index	Geographic Region of Journal
1st	*Journal of Pain & Symptom Management* (JIF = 3.61, Q2)	60 (8.9%)	360	10	North America
2nd	*Palliative Medicine* (JIF = 4.76, Q1)	32 (4.6%)	82	6	Europe
3rd	*American Journal of Hospice & Palliative Medicine* (JIF = 2.50, Q3)	30 (4.5%)	39	3	North America
4th	*Journal of Palliative Medicine* (JIF = 2.94, Q2)	22 (3.3%)	41	2	North America
5th	*Indian Journal of Palliative Care* (JIF = NA, Q4)	21 (3.1%)	33	3	India
6th	*Journal of Hospice & Palliative Nursing* (JIF = 1.91, Q3)	14 (2.1%)	24	2	North America
7th	*BMJ Supportive & Palliative Care* (JIF = 3.56, Q2)	11 (1.6%)	30	2	Europe
8th	*Soins* (JIF = NA, Q4)	11 (2.5%)	5	0	France
9th	*Palliative & Supportive Care* (JIF = 2.25, Q3)	6 (1.5%)	12	2	Europe
10th	*BMJ Open* (JIF = 2.69, Q2)	6 (1.5%)	14	3	Europe

JIF = journal impact factor; NA = not available. ^†^ Journal impact factor based on Thomson Reuters Web of Knowledge JCR Ranking (2020).

**Table 3 healthcare-10-01344-t003:** Top 10 productive countries on PC-related COVID-19 publications.

Rank	Country	Articles Numbers ^†^	No. of Citations	Citation Average	Percentage of Articles Published by the Country	COVID Cases (in Millions) ^‡^	Articles/Million COVID Cases	COVID Deaths (in 1000) ^‡^	Articles/1000 COVID Deaths
1st	USA	183	1902	10.39	27.2%	82.7	2.2	1.019	0.0001
2nd	United Kingdom	84	620	7.38	12.5%	21.9	3.8	174.1	0.48
3rd	India	45	77	1.71	6.7%	43.1	1.1	522.3	0.86
4th	Italy	39	289	7.41	5.8%	16.2	2.4	162.9	0.24
5th	Germany	35	150	4.28	5.2%	24.3	1.4	135.12	0.26
6th	Australia	31	131	4.22	4.6%	5.7	5.4	7.1	4.36
7th	France	26	206	7.92	3.9%	28.4	0.91	145.4	0.18
8th	Spain	26	129	4.96	3.9%	11.8	2.2	104.2	0.25
9th	Canada	23	129	5.61	3.4%	3.7	6.2	38.9	0.59
10th	Brazil	20	44	2.2	3.0%	30.3	0.6	662.9	0.03

^†^ The number of articles was counted based on the corresponding author’s country. ^‡^ World population estimate. Accessed on 27 April 2022 from https://www.worldometers.info/coronavirus/.

**Table 4 healthcare-10-01344-t004:** Most frequently used study keywords in PC-related COVID-19 research.

Rank	Keywords	Occurrences	Total Link Strength	Rank	Keywords	Occurrences	Total Link Strength
1st	COVID-19	291	381	21st	Critical care	5	7
2nd	Palliative care	193	311	22nd	Elderly	5	11
3rd	Cancer	23	49	23rd	Home care	5	10
4th	Telehealth	18	34	24th	Primary care	5	9
5th	Death and dying	16	35	25th	Prognosis	5	9
6th	Hospitalization	12	25	26th	Social work	5	13
7th	Bereavement	11	29	27th	Symptom management	5	11
8th	Communication	11	26	28th	Terminal care	5	14
9th	Advance care planning	10	20	29th	Treatment	5	12
10th	Nursing	10	19	30th	Adult	4	6
11th	Qualitative research	10	18	31st	Challenges	4	10
12th	Ethics	9	16	32nd	Dementia	4	10
13th	Quality of life	9	22	33rd	Education	4	6
14th	Grief	7	17	34th	Geriatrics	4	7
15th	Pediatric	7	10	35th	Goals of care	4	10
16th	Nursing homes	6	15	36th	Preparedness	4	7
17th	Palliative medicine	6	13	37th	Psychological distress	4	7
18th	Public health	6	13	38th	Radiotherapy	4	9
19th	Anxiety and depression	5	9	39th	Triage	4	8
20th	Chemotherapy	5	11				

## Data Availability

The data used in this study are available on reasonable request from the first and corresponding authors.

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
