# Peer review of "Palliative Care Landscape in the COVID-19 Era: Bibliometric Analysis of Global Research"

_healthcare, 2022, doi:10.3390/healthcare10071344_

Round 1

Reviewer 1 Report

Hammoda et al systematically searched four databases for the paper published during the last 2 yearsThey found Canada, Australia, and the United Kingdom were  the most productive countries regarding articles published per million confirmed COVID-19 cases by employing VOSviewer, Gephi and R software. They found a lack of international collaborations and uneven research focus on PC across countries with different pandemic trajectories. Research should expand to grief and bereavement support of families besides the diseases like cancer. The manuscript is generally informative and a few points should be addressed before publication.

1) The figures need to be revised. For instance, some of texts in Figure 1 are not shown very well.

2) The authors claimed they used VOSviewer, Gephi and R software to analyze the database. The pros and cons needs to be compared and shown in the manuscript.

Author Response

Dear Editor and reviewers,

Thank you for allowing us to revise our manuscript and to respond to the helpful comments raised by the reviewers. Below is a detailed response to comments and changes made to the manuscript.

Response to Reviewer (1)

Many thanks for your comments. All your comments were addressed in detail, as shown below.

Hammoda et al systematically searched four databases for the paper published during the last 2 years. They found Canada, Australia, and the United Kingdom were the most productive countries regarding articles published per million confirmed COVID-19 cases by employing VOSviewer, Gephi and R software. They found a lack of international collaborations and uneven research focus on PC across countries with different pandemic trajectories. Research should expand to grief and bereavement support of families besides the diseases like cancer. The manuscript is generally informative and a few points should be addressed before publication.

Comments (1): The figures need to be revised. For instance, some of texts in Figure 1 are not shown very well.

Response: All figures have been checked, and the texts in Figure 1 have been well-shown and adjusted to be visible.

Comments (2): The authors claimed they used VOSviewer, Gephi and R software to analyze the database. The pros and cons needs to be compared and shown in the manuscript.

Response: Sentences have been added in the section “Data analysis and visualization” to demonstrate the pros and cons of the software (Line 119-132, P 3).

Reviewer 2 Report

Relevant article.

It highlights the reality of Covid and its relationship with palliative care.

The relationship between Countries/Universities, where more articles were produced, and the characteristics of these articles/journals. It makes this reality visible.

My suggestion, and because it is an added value of the manuscript, is that the authors refer in the inclusion criteria that different languages were included. If they refer to the time limit, it also makes sense to refer to languages. Also, they mention that they do not include reviews, but editorials, etc., were also not included. This can be seen in the flow diagram but should be in the inclusion criteria to convey transparency in the process.

Lastly, you mention that two reviewers carried out the analysis process. Does it also make sense to say how you proceeded in case of non-consensus between the reviewers? Did you use a third reviewer?

Author Response

15th July. 2022

Manuscript ID: healthcare-1811620, entitled "Palliative Care Landscape in the COVID-19 Era: Bibliometric Analysis of Global Research"

Dear Editor and reviewers,

Thank you for allowing us to revise our manuscript and to respond to the helpful comments raised by the reviewers. Below is a detailed response to comments and changes made to the manuscript.

Response to Reviewer (1)

Many thanks for your comments. All your comments were addressed in detail, as shown below.

Hammoda et al systematically searched four databases for the paper published during the last 2 years. They found Canada, Australia, and the United Kingdom were the most productive countries regarding articles published per million confirmed COVID-19 cases by employing VOSviewer, Gephi and R software. They found a lack of international collaborations and uneven research focus on PC across countries with different pandemic trajectories. Research should expand to grief and bereavement support of families besides the diseases like cancer. The manuscript is generally informative and a few points should be addressed before publication.

Comments (1): The figures need to be revised. For instance, some of texts in Figure 1 are not shown very well.

Response: All figures have been checked, and the texts in Figure 1 have been well-shown and adjusted to be visible.

Comments (2): The authors claimed they used VOSviewer, Gephi and R software to analyze the database. The pros and cons needs to be compared and shown in the manuscript.

Response: Sentences have been added in the section “Data analysis and visualization” to demonstrate the pros and cons of the software (Line 119-132, P 3).

Response to Reviewer (2)

Many thanks for your comments. All your comments were addressed in detail as shown below.

It highlights the reality of Covid and its relationship with palliative care. The relationship between Countries/Universities, where more articles were produced, and the characteristics of these articles/journals. It makes this reality visible.

Comments (1): My suggestion, and because it is an added value of the manuscript, is that the authors refer in the inclusion criteria that different languages were included. If they refer to the time limit, it also makes sense to refer to languages. Also, they mention that they do not include reviews, but editorials, etc., were also not included. This can be seen in the flow diagram but should be in the inclusion criteria to convey transparency in the process.

Response: The inclusion and exclusion criteria have been adjusted based on your comments.

Lastly, you mention that two reviewers carried out the analysis process. Does it also make sense to say how you proceeded in case of non-consensus between the reviewers? Did you use a third reviewer?

 Response: Yes, another author has checked the discrepancies between the first two authors who extracted the data. A new sentence has been added “The third author (JB) was responsible for making the final decision of any uncertainty that the first two authors encountered during the assessment of the articles (Line 99-101, P 3).

Additional clarifications

In addition to the above comments, all spelling and gramatical errors pointed out have been corrected. The revised manuscript conforms to the journal style.

We are happy to provide any further clarification if necessary. We hope the above changes have now fully addressed the very useful comments made by the reviewers, and we appreciate their input and time on this, which helped us improve our paper. 

Sincerely,

Hammoda Abu-Odah

(on behalf of all authors)
